# Enhanced Structural, Optical, Electrical, and Dielectric Properties of PVA/Cu Nanocomposites for Potential Applications in Flexible Electronics

**DOI:** 10.3390/ma18092087

**Published:** 2025-05-02

**Authors:** Marco A. Alaniz Hernández, Carlos Ascencio Hurtado, Filiberto Candia Garcia, Roberto C. Ambrosio Lázaro, Manuel A. Chairez Ortega, Cesar A. Arriaga Arriaga, Amanda Carrillo Castillo

**Affiliations:** 1Manufacturing Department, Ciudad Juarez Autonomous University, Chihuahua 32310, Mexico; mhalaniz3@gmail.com; 2Electronic Department, Meritorious Autonomous University of Puebla, Puebla 72590, Mexico; carlos.ascenciohurtado.fce@viep.com.mx (C.A.H.); filiberto.candia@correo.buap.mx (F.C.G.); cesarau.arriaga@correo.buap.mx (C.A.A.A.); 3Electrical Department, Ciudad Juarez Autonomous University, Chihuahua 32310, Mexico; manuel.chairez@uacj.mx

**Keywords:** thin film technology, spin coating, copper nanoparticles, polymeric nanocomposites

## Abstract

Copper (Cu) nanoparticles, known for their high electrical conductivity and cost-effectiveness, have emerged as essential materials in various applications from flexible electronics to antimicrobial agents. This work focuses on the synthesis and characterization of semiconductive nanostructured films composed of polyvinyl alcohol (PVA) with embedded Cu nanoparticles. The study provides a comprehensive analysis of the structural, optical, electrical, and dielectric properties of the resulting nanocomposites. The results indicate a significant reduction in optical band gap, from 4.82 eV in pure PVA to 2.6–2.8 eV in the nanocomposites, alongside enhanced electrical conductivities reaching 1.20 S/cm for films with 5 wt.% Cu. Dielectric assessments further reveal high dielectric constants, underscoring the potential of these materials for flexible electronic applications. This work highlights the effectiveness of incorporating Cu nanoparticles into polymer matrices, paving the way for advanced materials that meet the demands of next-generation electronics.

## 1. Introduction

Nanotechnology has the potential to address urgent societal challenges through its diverse applications in fields such as information technology, energy production, environmental protection, biomedicine, and agriculture, among others [1,2]. The distinct properties exhibited by nanomaterials enable innovative solutions to contemporary manufacturing issues, primarily by reducing raw material consumption and enhancing the efficiency of applications, allowing for the use of fewer resources [2].

Copper (Cu) is one of the most essential electrical materials, primarily due to its high electrical conductivity—second only to silver—while being more abundant and relatively lower in cost [2]. Copper nanoparticles have demonstrated promising applications in diverse areas, including heat dissipation, antimicrobial and fungal agents, lubricants, metal injection molding, catalysts, flexible electronics, and transparent conductors, among others [3,4,5].

Copper nanoparticles (NPs) can be produced through various techniques, typically classified as bottom-up (chemical) and top-down (physical) methods. For the bottom-up approach, nanoparticles are assembled from atoms, molecules, or clusters. In contrast, top-down methods involve reducing a macroscopic material to nanometer dimensions via cutting, grinding, or milling without atomic-level control [6]. A third approach, biological synthesis, has emerged in recent years. Although it can be considered a subset of chemical methods, it presents unique challenges and opportunities. Physical techniques include thermal evaporation [7], laser ablation [8], and mechanical milling [9,10], while chemical synthesis methods encompass electrochemical deposition [11]; chemical reduction [12]; and photochemical [1], sonochemical [13], and polyol processes [14], among others.

On the other hand, thin film and coating technology, which are closely related, are techniques used to modify or enhance the properties of a substrate by applying a film or coating functionalized with nanomaterials [11]. Conductive films are typically deposited onto dielectric or less conductive substrates [15,16]. These films must retain their electrical, optical, and mechanical properties after deposition. Recently, metallic nanomaterials have been proposed to further improve the electrical and electronic properties of devices [11].

Polymeric compounds have been widely reported as host materials for various nanoparticle systems and are commonly used as stabilizing agents in different chemical environments, as they prevent the agglomeration and precipitation of particles despite their lower thermal stability [17]. Recently, these materials have gained significance due to their versatile applications, particularly in optoelectronic devices such as solar cells, light-emitting diodes, optical signal processing systems, and optical sensors [18]. Among the most used nanoparticles are metals such as Au, Ag, Co, and Cu [19]. However, the incorporation of other elements, such as Ti, Cr, Fe, and metal oxides, has also been explored. Ahmed et al. published a study on the synthesis of thin films from polyvinyl alcohol (PVA) nanocomposites containing copper nanoparticles and titanium oxide for energy storage applications. They reported that increasing the nanoparticle content within the polymer matrix enhanced the conductivity of the nanocomposites [20].

Conversely, Soliman and Vshivkov [21], in their development of nanocomposite films incorporating nanoparticles for optoelectronic applications, observed an increase in particle size after dispersing the nanoparticles within the polymer matrix. Similar to Ahmed’s findings [20], they concluded that the addition of metallic nanoparticles enhances the conductivity of nanocomposite films while reducing the optical bandgap as conductivity increases. Additionally, Joshi et al. [22] synthesized nanocomposite films incorporating silver and cobalt nanoparticles, focusing on their application in energy storage devices. Their study also highlighted the reduction in bandgap energy with the incorporation of metallic nanoparticles.

Copper films are of significant interest for energy applications due to their high electrical conductivity and excellent resistance to electromigration [6]. Various methods exist for preparing Cu films, with metal-organic chemical vapor deposition (MOCVD) being the most reported technique, owing to its ability to produce uniform films with high selectivity for desired elements [12]. However, solution-based deposition techniques (e.g., spin coating, dip coating, inkjet printing) also show potential, while electrodeposition can be effectively integrated into standard complementary metal oxide semiconductor (CMOS) processes and used to produce nanostructured layers [12,13]. This study aimed to synthesize semiconductive nanostructured films composed of polyvinyl alcohol (PVA) matrix with copper nanoparticles. We focused on obtaining and characterizing the electrical and structural properties of these films to evaluate their suitability as candidate materials for flexible electronics. Given the low temperature requirements of this approach, we anticipate that our findings will contribute to the development of innovative materials that meet the demands of next-generation electronic applications.

## 2. Materials and Methods

### 2.1. Reagents

The following reagents were used: copper sulfate (CuSO_4_) with a reactive grade of 99.99% (Sigma-Aldrich, St. Louis, MO, USA) as the metal precursor, ethylenediamine (C_2_H_8_N_2_) reagent grade 99% (Merck, Darmstadt, Germany) as a stabilizing agent, sodium borohydride (NaBH_4_) reagent grade 98.9% (Fermont, Lerma, Mexico) as the reducing agent, and deionized water (H_2_O) as the solvent. The method of synthesis was based on [23], only the stabilizing agent was changed from soybean extract to ethylenediamine and the pH was stabilized at 7 with pure water.

For the preparation of the polymeric matrix, polyvinyl alcohol (PVA) from Sigma Aldrich (Tlalnepantla de Baz, Mexico) with an average molecular weight (Mw) of 130,000 g/mol and 99% hydrolyzed was used, along with deionized water (H_2_O). The spin-coated conductive films were deposited on standard 25 mm × 75 mm glass and PET with indium tin oxide (ITO) substrates (surface resistivity of 100 Ω/sq, 1 ft × 1 ft × 7 mil sheet, Sigma Aldrich, St. Louis, MO, USA).

### 2.2. Synthesis of Copper Nanoparticles

For the synthesis of copper nanoparticles, a reaction setup was prepared with a hot plate, a beaker, and a magnetic stirrer. First, the temperature of the reaction medium (H_2_O) was brought to approximately its boiling point (~98 °C) under constant stirring (360 RPM). After stabilization, the copper source, anhydrous CuSO_4_, was added to the reaction medium and allowed to mix for 10 min.

Then, ethylenediamine was added, and after 10 min, the next reagent was added. Finally, the reduced agent, Na_2_BH_4_ 10% g/mL, was added and the reactor was removed from the heat plate. The reaction time after the addition of the reducing agent was 30 min.

To recover the solid copper nanoparticles, a vacuum filtration system was set up using a Kitazato flask, a Buchner funnel, filter paper, and a vacuum pump. The contents of the beaker were slowly decanted onto the filter paper until all solids were collected. Finally, the filter paper was sprayed with isopropyl alcohol and removed from the filtration system to be dried in a vacuum oven for 24 h at ~40 kPa.

### 2.3. Preparation of Polymeric Composite

PVA 5% solution was prepared by dissolving granular powder PVA in distilled water under heating at 80 °C and magnetic stirring for approximately 30 min. Then, the synthesized 5 wt.% Cu NPs were taken and mixed with the PVA solution. Subsequently, the sample was subjected to an ultrasonic bath process for 15 min and then to vortex agitation for another 10 min. Once the dispersion process was completed, the samples were stored in plastic Corning tubes for later use.

### 2.4. Synthesis of PVA/Cu Nanocomposite Films

The deposition process was carried out as follows: the PET ITO substrate was placed at the center of the spin coating equipment. The dimensions of the substrates were 3.75 (length) × 2.5 cm (width). A measure of 0.5 mL of the polymeric nanocomposite solution was applied to the substrate using a plastic pipette in a static deposition prior to starting the rotation. The spin speed varied from 500 RPM to 3000 RPM, while the deposition time was maintained at 60 s throughout. After deposition, the films were dried at 80 °C for 10 min. Finally, the number of layers deposited varied from 1 to 3. All deposits were performed under ambient laboratory conditions.

### 2.5. Instruments and Methods

The chemical characterization of the synthesized materials by Fourier transform infrared spectroscopy (FTIR) was performed using a Nicolet iS10 Thermo Scientific spectrophotometer (Waltham, MA, USA). The experiment was configured with OMNIC Lite standard software to record measurements in the 4000–400 cm^–1^ range, with the laboratory atmosphere and the diamond crystal of the ATR accessory as references.

UV-Vis spectroscopy was conducted using a Jenway 6850 UV-Vis spectrophotometer (East Moline, IL, USA), which provides wavelength resolution within the 300–1100 nm range. The samples were measured in dispersion form, with water used as the reference target, and in thin film form, where a clean substrate served as the reference.

With the optical characterization, the Tauc method was used to estimate the band gap value of the samples. The wavelength and the absorbance were used to calculate the transmittance of the samples with the following equation:(1)$$T\left(\%\right)=\frac{100}{10^{Absorbance}}$$

Then, the energy photon (*hv*) was calculated as follows:(2)$$hv=\frac{240}{Wavelength\;(nm)}$$

Also, the absorption coefficient was calculated as follows:(3)$$\alpha =\ln\left(\frac{100}{Transmittance}\right)$$

Finally, the square of the multiplication of α with hv was calculated. To obtain the band gap, the hv values were plotted on the X-axis, and then the (αhv)^2^ values were plotted on the Y-axis. The intersection with the X-axis was considered to be the band gap value [24].

Elemental and microstructural characterization were carried out using a JEOL JSM-7800F scanning electron microscope (Tokyo, Japan), with magnifications ranging from 25,000× to 100,000×.

For the microstructural analysis by X-ray diffraction (XRD), an “X’Pert MPD” Philips diffractometer was employed. Approximately 1.5 g of the sample was taken directly from the storage medium and placed on a sample holder located beneath the X-ray beam for measurement. For conductive films, the films were placed directly under the X-ray beam.

Surface analysis of the samples was performed using an MFP3D-SA atomic force microscope (AFM) from Asylum Research (Santa Barbara, CA, USA), with x, y scans up to 90 µm and Z scans up to 15 µm.

For the electrical characterization, current–voltage (I–V) measurements were conducted under dark conditions at room temperature to calculate the DC electrical conductivity (σ_dc_). First, PVA thin films were deposited onto ITO glass substrates. Then, a grid of electrically conductive Ag paint pads was applied and distributed evenly across the sample surface. Each pad was 2 mm in diameter and 1 µm thick, and they were spaced 5 mm apart. Following this, heat treatment was performed for 30 min at 90 °C, with a heating ramp of 5 °C/min and a cooling ramp of 2 °C/min. Current–voltage (I–V) measurements were then carried out using a Keithley 2450 electrometer (Beaverton, OR, USA). The room temperature conductivity (σ_RT_) was calculated using the following equation [25]:(4)$$\sigma_{dc}=\frac{I\cdot L}{V\cdot A}$$
where A is the area of the pads, L is the separation between pads, and I and V are the measured current and voltage, respectively. Additionally, frequency-dependent electrical properties were characterized using a Hioki IM 3536 LCR meter at room temperature (Nagano, Japan). Measurements of impedance (Z) and series capacitance (C_s_) were performed, allowing the calculation of the dielectric constant (ε) by applying a frequency sweep over the range of 10 Hz to 10^6^ Hz. The dielectric constant (ε) was calculated using Equation (2): ε = (C·d)/(ε_0_·A), where C is the series capacitance, d is the sample thickness, A is the electrode area, and ε_0_ is the vacuum permittivity constant. The LCR meter also accounted for cable length adjustments during the measurement process.

## 3. Results and Discussion

### 3.1. FTIR and UV-Vis of PVA Solution

Figure 1a shows the FTIR spectra of pure PVA. The broad band around 3475 cm^−1^ is attributed to the O–H stretching vibrations of PVA, while the absorption band at 1658 cm^−1^ is assigned to C=C stretching. The band at 1094 cm^−1^ is due to C-O stretching, corresponding to the carbonyl groups in the PVA backbone [26]. The band at 852 cm^−1^ corresponds to C–C stretching vibrations. Figure 1b shows the UV-Vis absorption spectrum of a 5% PVA solution. The absorption edge begins around 340 nm, consistent with the reported works on PVA [26,27], where the maximum absorption edge typically occurs between 270 and 200 nm [26,27]. A bandgap of 4.82 eV was calculated from the Tauc plot using this absorption spectrum [27,28].

### 3.2. Copper Nanoparticles

#### 3.2.1. FTIR and UV-Vis for Cu NPs

Figure 2 shows the FTIR spectrum of copper nanoparticles. For the measurements, a small number of nanoparticles were dispersed in the isopropyl alcohol used during the filtration process. The bands at 3150, 1680, 1431, and 1130 cm^−1^ correspond to O–H stretching vibrations (alcoholic), C–H asymmetric stretching, C–OH stretching, and C–OH bending vibrations, respectively. The band around 2400 cm^−1^ is attributed to the tensile motion of O=C=O bonds and is associated with CO_2_ from the atmosphere where the sample was taken, as no special atmospheric conditions were used for the measurements [29,30]. Additionally, the characteristic vibrational peak of Cu is observed at 613 cm^−1^ [31].

Figure 3a shows the UV-Vis absorption spectrum of the synthesized copper nanoparticles (NPs). The characteristic absorption maximum of Cu is observed at approximately 550 nm [31]. This peak corresponds to the surface plasmon resonance (SPR) of nanometer-sized copper particles, which has been reported in the literature [31,32].

According to the literature, it is possible to determine the size distribution of the nanoparticles obtained through the average amplitude of the absorption edge in the UV-Vis spectrum [33]. The larger the amplitude, the larger the size distribution in the sample, which means that nanoparticles of different sizes were obtained during the synthesis process, which can be seen in the SEM images (Figure 4). In the opposite case, i.e., if the average absorption edge width were smaller (narrower), the nanoparticle sizes would be more homogeneous.

The absorption coefficient (related to the absorption edge in the UV spectrum) and the band gap energy are connected through the Tauc equation. Thus, the band gap is determined by plotting (ahv)^2^ against hv (where α is the absorption coefficient, h is Planck’s constant, and v is the photon frequency). Using the UV-Vis spectrum data, the Tauc method was applied to calculate the band gap energy. Figure 3b shows the band gap plot for the copper nanoparticles, yielding a value of approximately 1.82 eV (represented by the intersection on the x axis of the black line). This value is consistent with the literature on copper nanoparticles, corresponding to the conductive behavior typical of this material.

Different studies have reported the synthesis of copper nanoparticles with band gap values in the range of 1.6–2.7 eV, arguing that obtaining a band gap above 2 eV is more attributable to the behavior of a semiconductor [34,35], which is explained by the difficulty of avoiding the oxidation of nanoparticles during the synthesis process. In these cases, the predominant material turns out to be copper oxide.

#### 3.2.2. SEM

Figure 4 shows the SEM micrographs (secondary electrons) of the copper nanoparticles, taken at magnifications of 10,000× and 40,000×, respectively. The images reveal that the nanoparticles predominantly exhibit a spheroidal morphology, with most particles measuring less than 100 nm in diameter. However, some particles exceed 100 nm in size, likely due to agglomeration. This agglomeration may be attributed to the presence of residual structural water molecules in the sample.

At higher magnifications, the larger particles observed are likely agglomerates of smaller particles, where spheroidal structures predominate. The particle sizes show significant variation, ranging from 59 nm to 112 nm, which can be attributed to the formation of these agglomerates. This agglomeration negatively impacts the desired properties of the nanomaterials by reducing their effective surface area.

The size and morphology of the synthesized nanoparticles are consistent with the results reported in the literature [36,37]. For example, in recent studies utilizing common synthesis methods, such as green synthesis, nanoparticle sizes have been reported to range from 20 nm to 80 nm [38,39]. In terms of morphology, spheroidal nanoparticles are typically observed, with occasional irregular structures caused by agglomeration.

#### 3.2.3. XRD

Figure 5 shows the XRD diffractogram of the copper nanoparticles, where the characteristic diffraction peaks of copper are observed at 2θ = 45° and 50°, corresponding to the (111) and (200) crystallographic planes, respectively [40]. These peaks are indicative of a face-centered cubic (FCC) structure. No secondary phase reflections were identified in the X-ray pattern. Additionally, the diffraction peaks, except for the one near 45°, are not sufficiently intense or sharp, suggesting a relatively low degree of crystallinity in the sample.

The intensity of the diffraction peaks is another essential feature to highlight in the diffractogram. A higher peak intensity generally indicates a larger crystallite size, as this characteristic can be directly related to nanoparticle size.

The low degree of crystallinity observed can be attributed to the absence of thermal treatment, which is often used to induce phase transitions and enhance crystallinity. Although thermal treatment can help achieve smaller crystallite sizes and induce specific phases, it may also promote the oxidation of the copper nanoparticles if not performed under a controlled atmosphere. This added complexity would not align with the objectives of this study. Nanoparticles with crystallite sizes ranging from 100 to 10 nm have been reported in the literature, but in all cases, the synthesis process includes a heat treatment above 200 °C [41].

### 3.3. PVA-Cu Nanocomposite Films

#### 3.3.1. UV-Vis

The UV-Vis absorption spectra of the PVA/Cu films with one, two, and three layers deposited at 500, 1000, 1500, and 3000 RPM over PET ITO are presented in Figure 6. All spectra exhibit a maximum absorption edge of copper nanoparticles at approximately 600 nm, with intensity increasing as the number of layers increases.

Absorption studies driven by surface plasmon resonance (SPR), a characteristic of noble metal nanoparticles (NPs), were conducted by analyzing the absorption spectra of the polymer matrix, Cu NPs, and the resulting nanocomposite. The peak at about 200 nm is attributed to PVA absorption, while the Cu NPs in the composite display broad plasmon absorption bands centered at approximately 600 nm (marked with red circles in the images). Notably, maximum absorption peaks for both PVA and Cu are observed in the films with two layers. This behavior may result from a better distribution of materials, leading to more-homogeneous films, which aligns with the objective of this research.

From the Tauc diagrams (representation shown in the inset of Figure 7 where band gap is represented by the intersection on the x axis of the black line), using the UV–visible absorption spectra, band gap values (Eg) of 2.8 eV, 2.6 eV, and 2.8 eV were calculated for films deposited with one, two, and three layers over PET ITO, respectively, as illustrated in Figure 7. These values are lower compared to those obtained for pure PVA [42,43]. Considering this analysis, it can be inferred that the results align with theoretical expectations, as the presence of a more defined absorption edge in the UV-Vis spectrum indicates a higher concentration of copper nanoparticles. Given that these nanoparticles have conductive characteristics, it is anticipated that the band gap will decrease as the concentration of the conductive material increases.

The band gap values for the two-layer and three-layer films are at their lowest when deposited at 500, 1000, and 1500 RPM. Beyond this speed, no favorable effect on the conductive characteristics of the material is observed. In contrast, the films deposited as a single layer exhibited the highest band gap values; however, the same trend in deposition speed was maintained.

#### 3.3.2. SEM

Figure 8, Figure 9 and Figure 10 show the SEM images of the polymer nanocomposite materials deposited over PET ITO after the incorporation of copper nanoparticles into the polymer matrix. A homogeneous distribution across the entire surface of the substrate is observed in all films, along with the formation of material clusters. The size and frequency of these clusters increase with both the deposition speed and the number of layers.

As the magnification increases, particles with a more well-defined spherical morphology become evident. Although some particles are still part of agglomerates, these agglomerates are smaller in size and exhibit a more clearly defined morphology.

Micrographs of films deposited over PET ITO in two layers at varying deposition speeds of 500, 1000, and 1500 RPM show the formation of larger aggregates. This effect is noticeable in the distribution of material on the surface of the substrate, where determined structures begin to grow from clusters or grains due to material accumulation.

Unlike single-layer films, the grain growth in two-layer films becomes more evident as the number of deposited layers increases. In general, a homogeneous distribution of material on the substrate is observed, along with the appearance of irregular morphologies generated by grain growth.

#### 3.3.3. XRD

Figure 11 shows the X-ray diffraction (XRD) patterns for two of the PVA-Cu nanocomposite films deposited with different numbers of layers. The characteristic diffraction peaks of copper are observed at 2θ = 45° and 50°, corresponding to the (111) and (200) planes, respectively. However, additional peaks are present at 38° (111), 60° (220), and 75° (220) [44]. Also, a peak appears at 19.5°, which corresponds to PVA. This is attributed to the semi-crystalline structure of PVA, due to intra- and intermolecular forces in its structure [45].

The latter is attributed to copper oxide nanoparticles, indicating the occurrence of oxidation either in the precursor solution or during the drying process due to thermal energy. In these samples, the effect of increasing the number of layers becomes more pronounced, as evidenced by the sharper definition of the diffraction peaks in the samples deposited with three layers.

Based on the described characterization results, it was determined that the samples deposited at 500 and 1000 RPM with two deposited layers exhibited desirable characteristics for application in the development of flexible electronic devices. Therefore, the following characterization focuses on highlighting the structural and electrical properties of these films.

#### 3.3.4. AFM

Figure 12 shows the AFM micrographs of the PVA-Cu nanocomposite films, where spherical structures of similar sizes are observed with a uniform distribution across the surface. The effect of grain boundary growth is evident in the AFM images. The film with the lowest roughness was the one deposited with two layers at 500 RPM, which can be attributed to the higher concentration of polymer in the sample. As indicated by the SEM images, the polymer appears to coat the copper particles more effectively at this deposition speed, contributing to the reduced roughness.

However, as the number of layers increased at 500 RPM, the roughness increased proportionally. This suggests that different interfaces may form between the layers rather than the new layers simply filling the valleys formed by the previous ones. Additionally, grain growth likely influences the roughness values, further contributing to the increase in surface roughness with more layers.

#### 3.3.5. I–V Curves

As a function of the applied voltage (V), the current (I) flowing through the doped copper PVA thin films was measured. Figure 13 illustrates the I–V characteristics of PVA/Cu (5%) with two layers of deposited films for the two deposition speeds (500 and 1000 RPM). The current (I) increases with applied voltage (V), initially at a slow rate, followed by a more rapid increase, indicating non-Ohmic behavior. This variation in the I–V characteristics exhibits semiconducting behavior. These parameters were calculated with Equation (4).

Table 1 presents a comparison of DC conductivity values for pure PVA and PVA/Cu nanocomposites, with the last three rows showing the results obtained in this study. A substantial increase in DC conductivity is observed for the PVA/Cu nanocomposites synthesized at different deposition speeds, particularly when compared with the published results for both pure PVA and nanocomposites with a lower Cu content.

The results from this study demonstrate significantly higher DC conductivities for PVA/Cu (5 wt.%) nanocomposites, with values of 1.04 S/cm and 9.40 × 10^−1^ S/cm at 500 RPM, and 1.20 S/cm at 1000 RPM. These values are several orders of magnitude higher than the highest reported conductivity for PVA/Cu nanocomposites (1.27 × 10^−2^ S/cm). This significant enhancement in conductivity can be attributed to three key factors: higher Cu content, higher deposition speed, and improved interfacial polarization. First, the 5 wt.% Cu content in these samples introduces a greater concentration of conductive nanoparticles, enabling the formation of more-effective conductive networks throughout the polymer matrix. Second, the deposition speeds of 500 and 1000 RPM may have contributed to the formation of more-uniform films with better nanoparticle dispersion, leading to improved charge mobility. Lastly, the strong interaction between the Cu nanoparticles and the PVA matrix enhances interfacial polarization, further boosting the overall conductivity of the nanoparticles.

Therefore, the increase in DC conductivity observed in this work for PVA/Cu nanocomposites, particularly at 5 wt.% Cu and higher deposition speeds, underscores the effectiveness of the fabrication methods employed. These results represent a considerable advancement over nanocomposites reported in the literature and suggest that the nanocomposites synthesized here hold significant potential for applications requiring high electrical conductivity, such as flexible electronics, conductive coatings, and energy storage devices.

Figure 14 presents the impedance (Z) versus frequency (Hz) behavior of a PVA/Cu nanocomposite sample containing two layers of PVA/Cu (5%) synthesized at two different deposition speeds (500 RPM and 1000 RPM). The curves exhibit typical behavior for dielectric materials, where the impedance decreases as the frequency increases. This trend is due to the capacitive nature of the material, as at higher frequencies, the ability of dipoles to align with the alternating electric field diminishes, leading to a reduction in impedance. At lower frequencies (around 10 Hz), both 500 RPM samples (represented by black and blue curves) follow similar trends, with the blue curve (med-6.1x) showing slightly higher impedance throughout the frequency range than the black curve (med-1x). This indicates that slight variations in measurement or film preparation affect the dielectric response, with the blue curve representing more-resistive material. Meanwhile, the 1000 RPM sample (red curve) exhibits the lowest impedance across all frequencies, indicating better electrical conductivity.

The contrast between the blue and red curves highlights the significant influence of deposition speed on the material’s impedance. A slower deposition rate (500 RPM) appears to produce a more resistive structure, which may be due to factors such as increased film thickness or less-effective formation of conductive pathways. Both deposition speeds show typical dielectric relaxation, where impedance is higher at lower frequencies due to polarization effects within the polymer matrix. At these lower frequencies, the dipoles have sufficient time to align with the external field, resulting in higher impedance. As the frequency increases, the dipoles and charge carriers within the material are less able to align with the alternating electric field, reducing the polarization contributions and resulting in lower resistance.

On the other hand, the sharp decrease in impedance at lower frequencies for the 1000 RPM sample suggests better charge mobility or improved conductive networks within the polymer matrix. The higher speed may have led to more-uniform Cu nanoparticle distribution, enhancing the electrical pathways. In contrast, the 500 RPM sample shows higher impedance, possibly due to thicker films or less-optimal nanoparticle dispersion, leading to greater resistance.

Figure 15 illustrates the capacitance (Cs) as a function of frequency (Hz) for PVA/Cu (5%) nanocomposite films with two layers deposited, synthesized at two different deposition speeds (500 RPM and 1000 RPM). The curves reveal how capacitance changes across a frequency range from 10 Hz to 1 MHz.

As in the previous graph (impedance), there is a reduction in the magnitude of the capacitance as the frequency increases for all samples, showing the same trend for dielectric materials. Also, at lower frequencies, the polarization mechanisms (such as dipolar alignment) have enough time to respond to the alternating electric field, resulting in higher capacitance. As the frequency increases, polarization cannot keep up with the fast-changing field, leading to a reduction in capacitance.

At low frequencies, the capacitance reaches values on the order of 10^−7^ F for the 1000 RPM sample, indicating strong polarization effects. In contrast, the 500 RPM samples show lower capacitance values (around 10^−9^ F), suggesting less-effective polarization, possibly due to differences in film homogeneity or nanoparticle distribution. Meanwhile, at higher frequencies (closer to 1 MHz), all curves converge to similar lower capacitance values, indicating that at very high frequencies, the material’s ability to store charge diminishes uniformly regardless of the deposition speed. Hence, the capacitance behavior of the PVA/Cu nanocomposites is strongly influenced by both the frequency of the applied field and the deposition speed. Higher deposition speeds (1000 RPM) enhance the capacitance across the frequency range, likely due to better nanoparticle dispersion and improved film quality, making the material more effective at storing charge. This suggests that deposition speed is a critical factor in optimizing the dielectric properties of PVA/Cu nanocomposites.

Figure 16 shows the dielectric constant (ε) as a function of frequency (Hz) for PVA/Cu (5%) nanocomposite films with two deposited layers synthesized at two different deposition speeds (500 RPM and 1000 RPM). The dielectric constant (dimensionless) is plotted over a frequency range from 10 Hz to 1 MHz. Again, as in the previous graphs, the behavior of a dielectric material was evident, since for all samples, the dielectric constant decreases as the frequency increases.

At lower frequencies (10–100 Hz), the dielectric constant reaches high values up to 10^5^ for the 1000 RPM sample, indicating strong polarization effects and significant charge storage capacity. As the frequency increases, the dielectric constant decreases, reflecting the decreased polarization response. The red curve, representing the 1000 RPM deposition speed, consistently shows higher dielectric constant values across the entire frequency range compared to the blue and black curves, which correspond to the 500 RPM deposition speed. This suggests that the higher deposition speed (1000 RPM) leads to a more homogeneous dispersion of Cu nanoparticles, enhancing interfacial polarization and improving the overall dielectric properties. In contrast, the blue and black curves, both representing samples produced at 500 RPM, show lower dielectric constant values. This is likely due to less-effective nanoparticle dispersion and potentially thicker or less uniform films, which reduce the material’s polarization capacity.

As the frequency approaches the megahertz range, all samples display a significant decrease in dielectric constant, converging to values between 10^2^ and 10^3^. This behavior is typical of dielectric relaxation phenomena, where the material’s ability to store charge diminishes because the dipoles cannot realign quickly enough with the rapidly oscillating electric field. Therefore, these results confirm that the electrical properties of PVA/Cu nanocomposites are highly dependent on both the frequency and the deposition speed, with higher deposition speeds leading to better dielectric performance due to improved nanoparticle dispersion and film quality.

In Table 2, the dielectric constant (ε) values for pure PVA and PVA/Cu nanocomposites at different frequencies (100 Hz to 100 kHz) are presented. The first seven rows represent results from studies reported in literature, while the last three rows present findings from this work. A comparison of these results reveals significant enhancements in the dielectric properties of the PVA/Cu nanocomposites synthesized in this study, particularly at lower frequencies. This highlights the remarkable impact of processing parameters, such as deposition speed and the higher concentration of Cu nanoparticles (5% wt.), on the dielectric properties.

For the first sample, the dielectric constant reaches 280.38 at 100 Hz, more than eight times higher than the value reported by Mohammed et al. (2022) [49] for 4 wt.% Cu (34.37). This dramatic enhancement is likely due to the improved dispersion and distribution of Cu nanoparticles at this deposition speed, which enhances interfacial polarization and charge storage capacity. At 1 kHz, the dielectric constant decreases to 109.27, but it remains substantially higher than the reported values for both pure PVA and PVA/Cu nanocomposites.

For the second sample, a similar trend is observed, with values of 214.73 at 100 Hz and 116.46 at 1 kHz. While these values are slightly lower than those of the first sample, they are still significantly higher than the literature values, emphasizing both the reproducibility and the strong impact of the synthesis method used in this study. For the third sample, the dielectric constant at 100 Hz reaches an exceptionally high value of 58,004.1, which is far beyond any previously reported value. This striking enhancement is likely due to a synergistic effect caused by the increased deposition speed, higher Cu nanoparticle content, and nanostructuring, leading to stronger interfacial polarization between the copper and PVA matrix. Even at 100 kHz, the dielectric constant remains relatively high at 1438.43, demonstrating excellent dielectric performance over a wide frequency range.

The addition of 5 wt.% Cu significantly enhances the dielectric properties compared to those of the studies that used a lower Cu content (e.g., 0.09% and 4% wt. in [47,49]). The larger volume of Cu nanoparticles leads to a greater surface area available for polarization, thus improving the overall conductivity and dielectric response of the films. Deposition speed also plays a critical role in the dielectric properties. At 500 RPM, the dielectric constant is substantially enhanced due to better Cu nanoparticle dispersion and the formation of more-homogeneous films. However, at 1000 RPM, the dielectric constant reaches extraordinarily high values, which may be attributed to the formation of conductive pathways within the nanocomposite, further enhancing its capacitance and polarizability. As expected, the dielectric constant decreases with increasing frequency due to the inability of dipoles to reorient rapidly enough in response to the oscillating electric field. However, even at higher frequencies (100 kHz), the dielectric constants of the samples from this work remain higher than those reported in the literature for pure PVA and PVA/Cu composites with lower Cu concentrations.

In summary, the results obtained in this work demonstrate a substantial improvement in the dielectric properties of PVA/Cu nanocomposites compared to previous studies. The combination of higher Cu nanoparticle content and optimized deposition speed leads to enhanced interfacial polarization and charge storage capacity, resulting in much higher dielectric constants, particularly at lower frequencies. These findings suggest that the PVA/Cu nanocomposites synthesized under these conditions could be highly suitable for applications that require high dielectric constants, such as flexible electronic devices and energy storage systems.

## 4. Conclusions

This study successfully synthesized and characterized nanostructured films composed of polyvinyl alcohol (PVA) with embedded copper (Cu) nanoparticles. The deposition was homogenous when using the spin coating method, maintaining these characteristics across one, two and three layers at 500, 1000, 1500 and 3000 RPM of varying deposition.

The FTIR analysis confirmed the presence of functional groups indicative of the successful incorporation of Cu nanoparticles within the PVA matrix, with characteristic peaks corresponding to both the polymer and the nanoparticles. The SEM images revealed a homogeneous distribution of spherical Cu nanoparticles, with increased aggregation observed at higher deposition speeds and layer counts, highlighting the influence of processing parameters on film morphology. The UV-Vis results demonstrated a decrease in the optical band gap from 4.82 eV for pure PVA to 2.6–2.8 eV for PVA/Cu nanocomposites, values corresponding to a semiconductive material. The observed reduction in band gap correlates with an increase in copper nanoparticle concentration, suggesting enhanced charge carrier mobility due to improved conductive networks within the matrix.

Electrical conductivity measurements indicated significantly enhanced DC conductivities, reaching values of 1.20 S/cm for films deposited at 1000 RPM with 5 wt.% Cu. This substantial increase can be attributed to effective nanoparticle dispersion, enhanced interfacial polarization, and the formation of conductive pathways within the polymer matrix.

Dielectric property assessments revealed remarkable performance, particularly at lower frequencies, with dielectric constants reaching values far exceeding those reported in the existing literature. These electrical properties, in combination with the band gaps, confirmed the semiconductive nature of the films. The results indicated that higher deposition speeds and increased Cu content significantly enhanced the dielectric response, making these nanocomposites suitable material for flexible electronic applications and energy storage devices.

## Figures and Tables

**Figure 1 materials-18-02087-f001:**
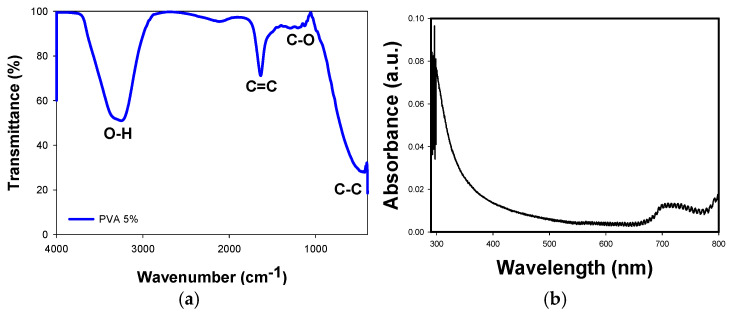
(**a**) FTIR spectra of PVA, (**b**) UV-VIS absorbance spectra of PVA.

**Figure 2 materials-18-02087-f002:**
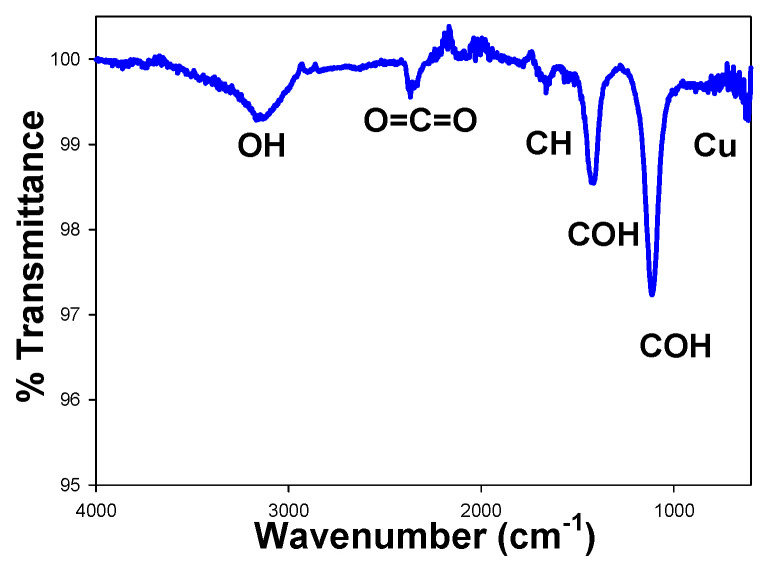
FTIR spectra of Cu NPs.

**Figure 3 materials-18-02087-f003:**
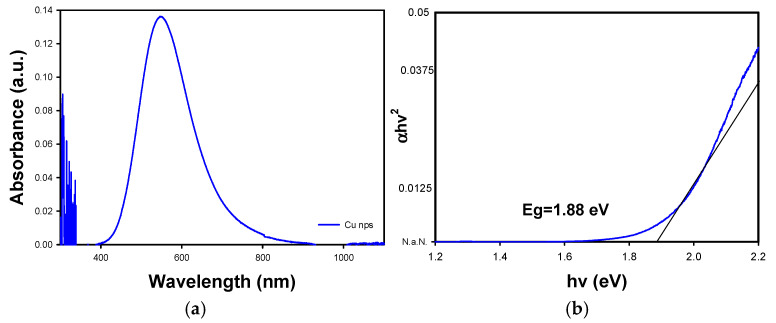
(**a**) UV-Vis absorbance spectra of Cu NPs, (**b**) Tauc variable plotted against energy for Cu NPs.

**Figure 4 materials-18-02087-f004:**
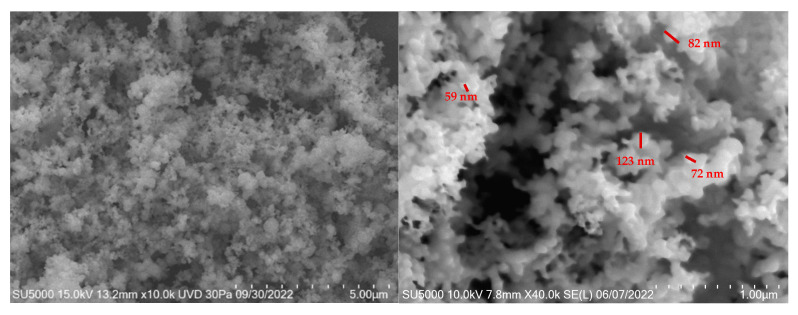
SEM micrographs for Cu NPs.

**Figure 5 materials-18-02087-f005:**
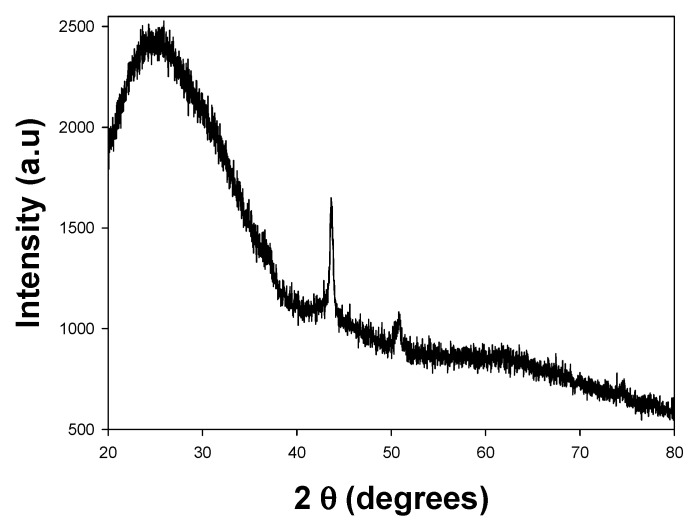
XRD pattern for Cu NPs.

**Figure 6 materials-18-02087-f006:**
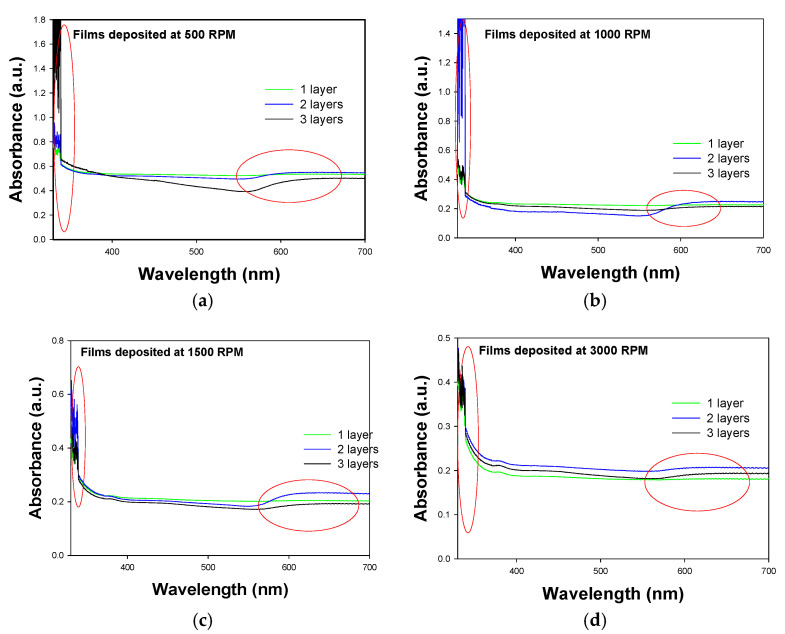
UV-vis absorbance spectra of PVA/Cu nanocomposite films deposited over PET ITO with different numbers of layers at different speeds: (**a**) 500 RPM (**b**) 1000 RPM (**c**) 1500 RPM (**d**) 3000 RPM.

**Figure 7 materials-18-02087-f007:**
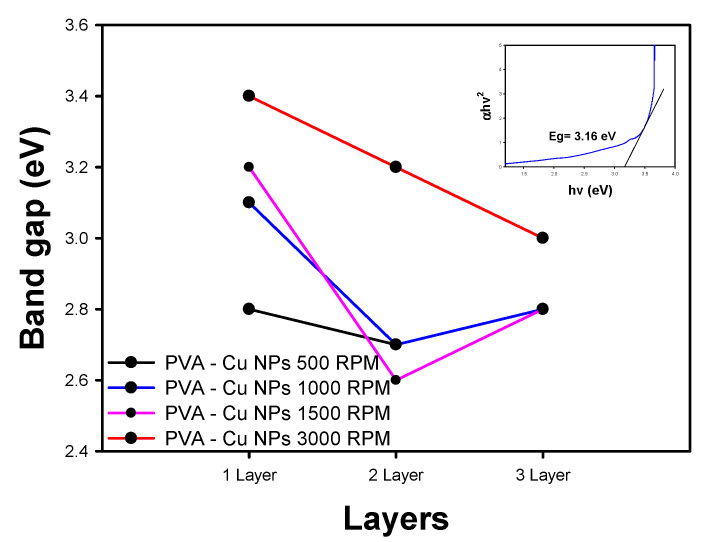
Band gap determination for PVA/Cu NP films deposited over PET ITO.

**Figure 8 materials-18-02087-f008:**
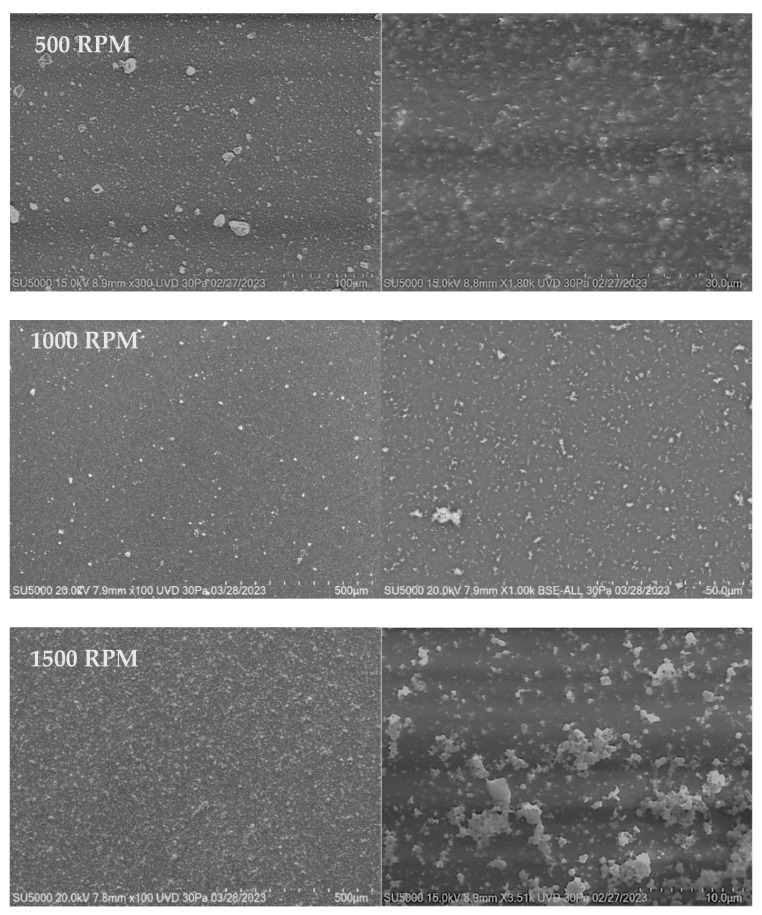
SEM micrographs for PVA-Cu NP films deposited over PET ITO at different speeds for one layer.

**Figure 9 materials-18-02087-f009:**
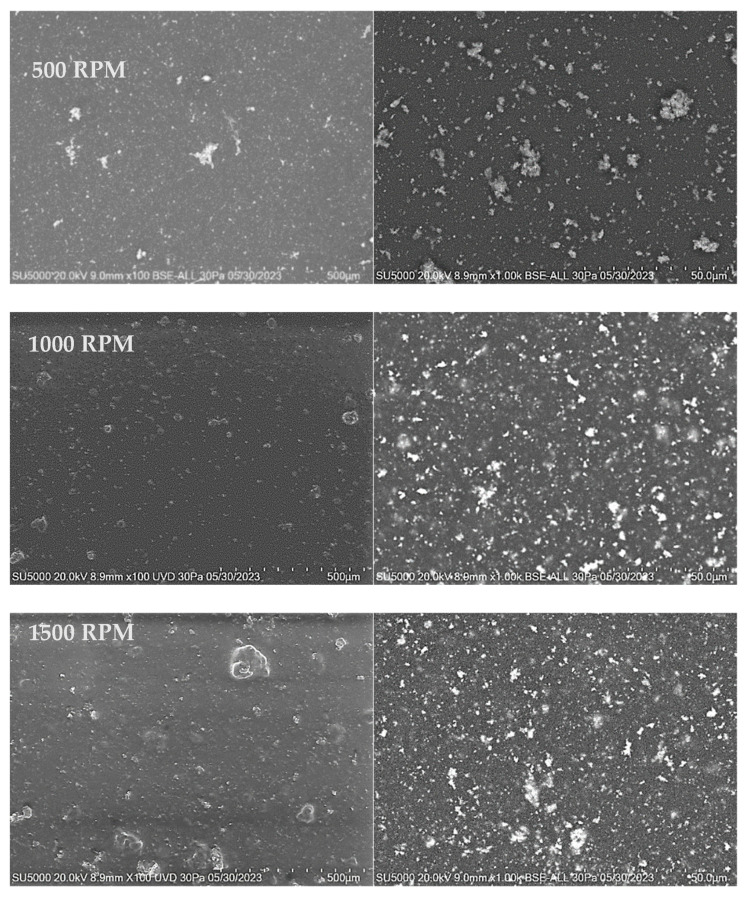
SEM micrographs for PVA-Cu NP films deposited over PET ITO at different speeds for two layers.

**Figure 10 materials-18-02087-f010:**
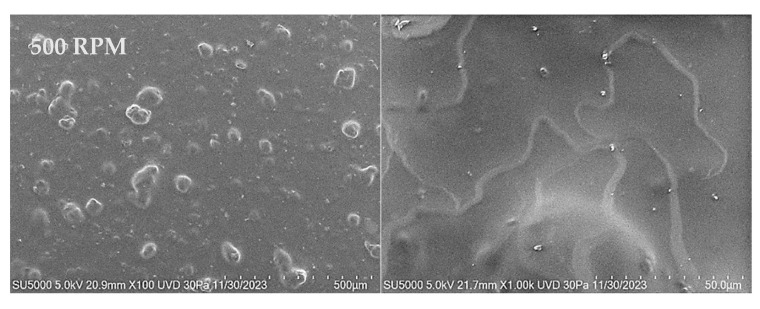
SEM micrographs for PVA-Cu NP films deposited over PET ITO at different speeds for three layers.

**Figure 11 materials-18-02087-f011:**
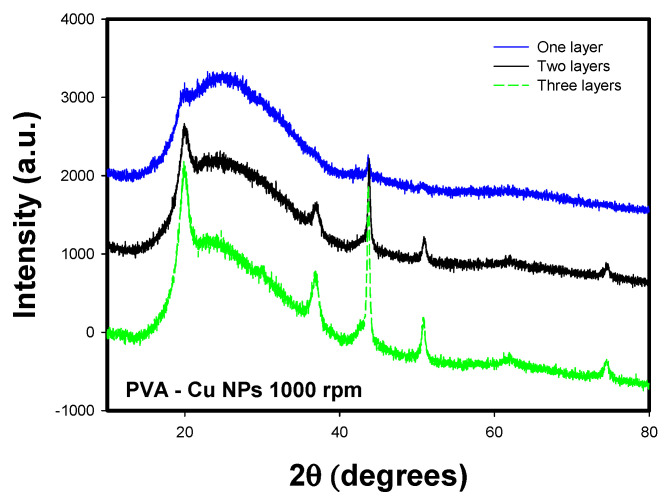
XRD patterns for PVA-Cu NP films deposited over PET ITO.

**Figure 12 materials-18-02087-f012:**
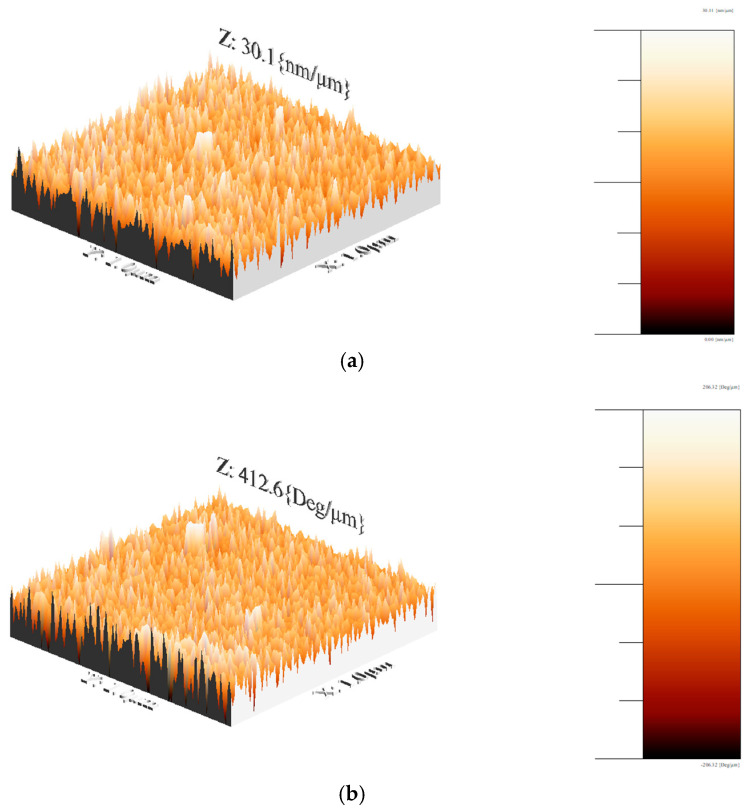
Three-dimensional AFM surface topography for PVA/Cu nanocomposite films deposited over PET ITO in (**a**) two layers (500 RPM) with an average roughness of 5 nm and (**b**) two layers (1000 RPM) with an average roughness of 73.7 nm.

**Figure 13 materials-18-02087-f013:**
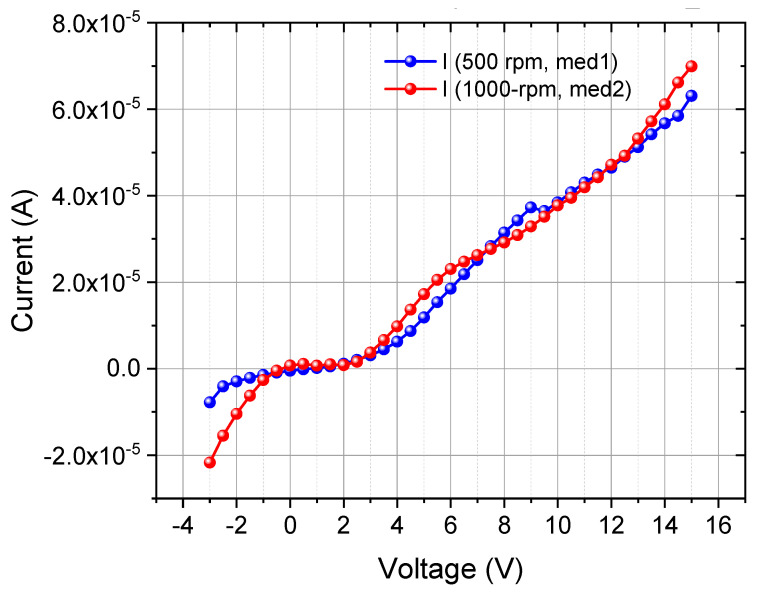
I–V curves for PVA/Cu (5%) nanocomposite films deposited over PET ITO in two layers.

**Figure 14 materials-18-02087-f014:**
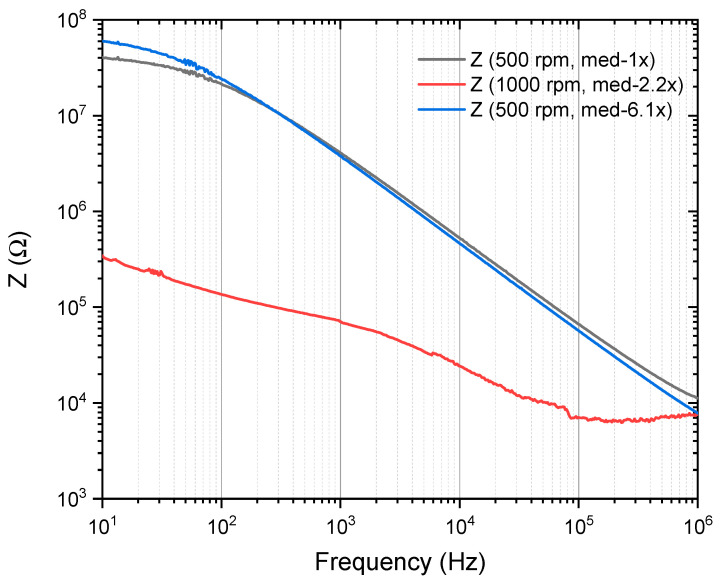
Impedance (Z) evolution for PVA/Cu (5%) nanocomposite films deposited over PET ITO at two layers throughout frequency evaluation.

**Figure 15 materials-18-02087-f015:**
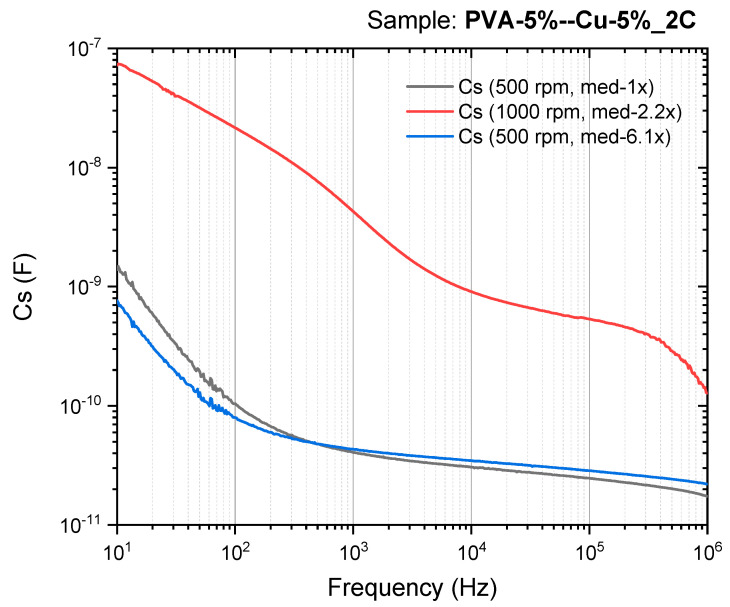
The series capacitance (Cs) for PVA/Cu (5%) nanocomposite films deposited over PET ITO in two layers.

**Figure 16 materials-18-02087-f016:**
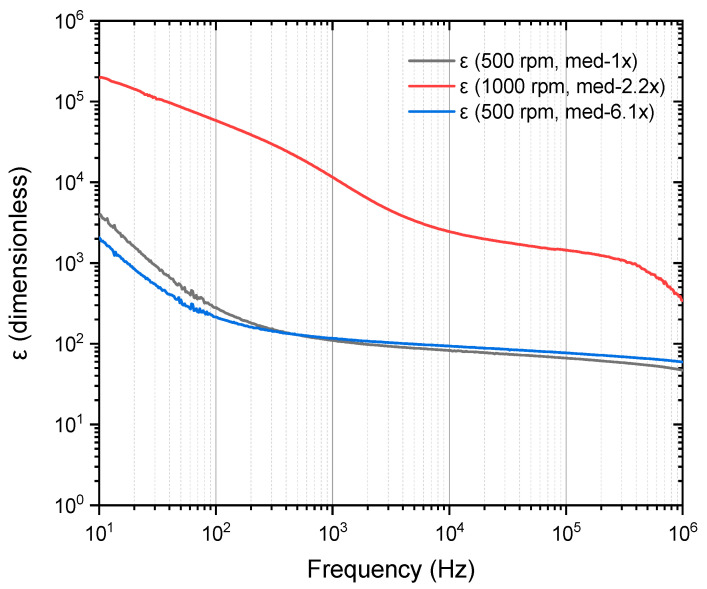
The calculated dielectric constant (ε) for PVA/Cu (5%) nanocomposite films deposited over PET ITO in two layers.

**Table 1 materials-18-02087-t001:** Electrical conductivity results for pure PVA and PVA-Cu nanocomposite films deposited over PET ITO in two layers.

Sample	DC Conductivity (S/cm)	Reference
Pure PVA	1.38 × 10^−11^	[46]
Pure PVA	3.70 × 10^−7^	[47]
PVA/Cu (0.09% wt.)	5.80 × 10^−10^	[48]
PVA/Cu (5% wt.)	1.27 × 10^−2^	[49]
PVA-ZnS/Cu (5%)	2.60 × 10^−9^	[50]
PVA/Cu (5%), 500 RPM	1.04	This work
PVA/Cu (5%), 500 RPM	9.40 × 10^−1^	This work
PVA/Cu (5%), 1000 RPM	1.20	This work

**Table 2 materials-18-02087-t002:** Dielectric constant (ε) results for pure PVA and PVA-Cu samples.

Sample	Dielectric Constant (ε)				Reference
	100 Hz	1 kHz	10 kHz	100 kHz	
Pure PVA	5	2	-	-	[46]
Pure PVA	--	--	--	2.38	[47]
Pure PVA	9.1	8.0	7.4	--	[48]
Pure PVA	19.66	17.8	--	--	[49]
Pure PVA	--	14.73	--	--	[50]
PVA/Cu (0.09% wt.)	--	--	--	1.19	[47]
PVA/Cu (4% wt.)	34.37	31.3	--	--	[49]
PVA/Cu (5%), 500 RPM	280.38	109.27	82.50	66.21	This work
PVA/Cu (5%), 500 RPM	214.73	116.46	93.52	76.82	This work
PVA/Cu (5%), 1000 RPM	58,004.1	11,453.4	2441.01	1438.43	This work

## Data Availability

The original contributions presented in this study are included in the article. Further inquiries can be directed to the corresponding authors.

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
