# Peer review of "Enhanced Structural, Optical, Electrical, and Dielectric Properties of PVA/Cu Nanocomposites for Potential Applications in Flexible Electronics"

_materials, 2025, doi:10.3390/ma18092087_

Round 1
Reviewer 1 Report
Comments and Suggestions for Authors
Dear authors,
After reviewing the manuscript, I have identified several significant issues that prevent me from recommending it for publication.
Best regards,

Author Response
Response for Reviewer 1
First of all, we would like to thank the Editori and the reviewers for all their comments and suggestions on the paper. We have corrected and completed the paper to address their comments and suggestions.
Comment: P4 L185: Could the authors present Figure 1b) in the wider range, from 300 to 1100 nm, to clearly shows that the absorption edge begins around 300 nm.
Response: Done. A graph with more important signals is now shown.
Figure 1. (a) FTIR spectra for PVA, (b) UV-VIS absorbance spectra of PVA.
Comment: It is unclear how the bandgap was estimated to be 3.42.eV. In the references cited by the authors [24, 25], values of bandgap are significantly different (about 6.18eV [24] and 6.2eV [25] for pure PVA). It would be significant to explain these differences.
Response: The band gap was recalculated. An additional reference was added.
“A bandgap of 4.82 eV was calculated from the Tauc plot using this absorption spec-trum [26, 27].”
[26] Nimrodh Ananth, A; Umapathy, S; Sophia, J; Mathavan, T; Mangalaraj D. On the optical and thermal properties of in situ/ex situ reduced Ag NP’s/PVA composites and its role as a simple SPR-based protein sensor. Appl Nanosci. 2011, 1. https://doi.org/10.1007/s13204-011-0010-7
Comment: P5 L200: In text of manuscript, paragraph 3.2.1. The authors is observed peak 613cm-1 and stated that is consistent with the UV-Vis results. Please, this needed explanation.
Response: We referred to the consistent formation of copper nanoparticles demonstrated by FTIR and UVVIS. To avoid confusion, we separated the descriptions.
“Additionally, the characteristic vibrational peak of Cu is observed at 613 cm−1 [30].
Figure 3a shows the UV-Vis absorption spectrum of the synthesized copper nano-particles (NPs). The characteristic absorption maximum of Cu is observed at approxi-mately 550 nm [30]. This peak corresponds to the surface plasmon resonance (SPR) of nanometer-sized copper particles, which has been reported in the literature [30,31].”
Comment: It would be clearer to use arrows for peaks on the FTIR spectra and also to show the corresponding functional groups (Figure1, Figure2a).
Response: Figure 1 was edited with the functional groups of the material.
Figure 1. (a) FTIR spectra for PVA, (b) UV-VIS absorbance spectra of PVA.
Figure 2 was edited with the functional groups of the material.
Figure 2. FTIR spectra for Cu NPs.
Comment: P5 Figure 3: Figure 3b) is not adequately presented. For example; the labels on the y-axis, values, magnitude, Eg-value.
Response: The axis was corrected in the plot.
Figure 3. a) UV-Vis absorbance spectra of Cu NPs, b) Tauc variable versus Energy plot for Cu NPs.
Comment: P8. L288: There is no inset of Figure 7.
Response: The inset was added.
Figure 7. Band gap determination for PVA-Cu NPs films.
Comment: P11: L346. Figure 11: One can notice a peak around of 2θ=18°. The authors should provide an explanation for this.
Response: We added the following explanation with a reference that supports this discussion:
“Also, a peak appears at 19.5°, which corresponds to PVA. This is attributed to the semi-crystalline structure of PVA, due to intra ant intermolecular forces in its structure [43].”
[43] Aziz, S. B.; Abdulwahid, R. T.; Rasheed, M. A.; Abdullah, O. G.; Ahmed, H. M. Polymer blending as a novel approach for tuning the SPR peaks of silver nanoparticles. Polym. 2017, 9, 486. http://dx.doi.org/10.3390/polym9100486
Comment: I noticed a lot of typographical errors in the text. It is necessary to carefully check and correct through the text of manuscript. Some of them are listed: Page3, L105: Instead of H2O should be written H2O, L116: Instead of Na2BH4 should be written Na2BH4, L93, L145: There is a missing period in the text, P5, L197: CO2, P15 L456: 10-7, L458: 10-9, P16 L479: 105 L491 102,103
Response: The indicated typographical errors were corrected, and also others typo errors detected were attended.
“hydrolyzed was used, along with deionized water (H2O). The spin-coated conductive films”
“Finally, the reduced agent, Na2BH4 10% g/mL, was added and the reactor was removed”
“materials that meet the demands of next-generation electronic applications.
spectrophotometer, which provides wavelength resolution within the 300 nm–1100 nm range.”
“of O=C=O bonds and is associated with CO2 from the atmosphere where the sample was”
“At low frequencies, the capacitance reaches values on the order of 10−7 F for the 1000
lower capacitance values (around 10−9 F), suggesting less effective polarization, possibly
105 for the 1000 rpm sample, indicating strong polarization effects and significant charge
decrease in dielectric constant, converging to values between 102 to 103.”

Reviewer 2 Report
Comments and Suggestions for Authors
see attachment

Author Response
Response for Reviewer 2
First of all, we would like to thank the Editori and the reviewers for all their comments and suggestions on the paper. We have corrected and completed the paper to address their comments and suggestions.
Comment: In the Introduction chapter, it is recommended to give the reference in line 71 (Conversely, Soliman and Vshivkov [ref]….); at line 73, (Similar to Ahmed's et al [ref]…); at line 76 (Additionally, Joshi et al [ref]..)
Response: The paragraph was modified adding the references in the indicated lines:
“Conversely, Soliman and Vshivkov [21], in their development of nanocomposite films incorporating nanoparticles for optoelectronic applications, observed an increase in particle size after dispersing the nanoparticles within the polymer matrix. Similar to Ahmed's findings [20], they concluded that the addition of metallic nanoparticles en-hances the conductivity of nanocomposite films while reducing the optical bandgap as conductivity increases. Additionally, Joshi et al [22], synthesized nanocomposite films incorporating silver and cobalt nanoparticles, focusing on their application in energy storage devices. Their study also highlighted the reduction in bandgap energy with the incorporation of metallic nanoparticles.”
[21] Soliman, T. S.; Vshivkov, S. A. Effect of Fe nanoparticles on the structure and optical properties of polyvinyl alcohol nano-composite films. J. Non Cryst. Solids 2019, 519. doi: 10.1016/j.jnoncrysol.2019.05.028.
[22] Joshi, A.; Mukherjee, G. S.; Banerjee, M.; Gupta, M. Effect of Ag underlayer on structural and optical properties of PVA/Ag/Co film. AIP Conf. Proc. 2020, 2220. doi: 10.1063/5.0001419.
Comment: 2.1. Reagents (Materials, it is recommended); „For the synthesis of Cu nanoparticles, it was based on [23]” - it is recommended to make it concrete; Also the sentence from lines 105-108, with paragraph and rephrased - the paper probably doesn't talk about PET
Response: The role of the stabilizing agent was declared, making concrete the adjustment of the methodology.
“The following reagents were used: copper sulfate (CuSO4) with a reactive grade of 99. 99% (Sigma-Aldrich, USA) as the metal precursor, ethylenediamine (C₂H₈N₂) rea-gent grade 99% (Merck, Germany) as a stabilizing agent, sodium borohydride (NaBH4) reagent grade 98.9% (Fermont, Mexico) as the reducing agent, and deionized water (H2O) as the solvent. The method of synthesis was based on [23], only changing the stabilizing agent from soybean extract to ethylenediamine and stabilizing the pH at 7 with pure water.”
Comment: In 2.2. subchapter, the sentences from lines 119-123 the term, copper nanoparticles, should specifically appear
Response: The term copper nanoparticles was added to the sentence missing it.
“To recover the solid copper nanoparticles, a vacuum filtration system was set up using a Kitazato flask, a Buchner funnel, filter paper, and a vacuum pump.”
Comment: Subchapter 2.4 is recommended to be reformulated, what substrate was used, the dimensions of the substrate, how the amount of 0.5 ml was dosed if the rotation time was 60s
Response: We added the requested information to the indicated paragraph:
“The deposition process was carried out as follows: the PET ITO substrate was placed at the center of the spin coating equipment. The dimensions of the substrates were 3.75 (length) x 2.5 cm (width). 0.5 mL of the polymeric nanocomposite solution was applied to the substrate using a plastic pipette in a static deposition, prior to starting the rotation. The spin speed varied from 500 RPM to 3000 RPM, while the deposition time was kept always at 60 seconds. After deposition, the films were dried at 80 ⁰C for 10 minutes. Finally, the number of layers deposited varied from 1 to 3. All deposits were performed under ambient laboratory conditions.”
Comment: Subchapter 2.5, the paragraph from lines 140-143 is recommended to be reformulated so that the term FTIR appears; line 146 - what solvents were used?; the method of making the electrodes is presented but the nature of the electrodes is not specified; references to equations are recommended;
Response: We added the requested information in the following pharagraphs:
“The chemical characterization by Fourier transform infrared spectroscopy (FTIR) of the synthesized materials was performed using a Thermo Scientific spectrophotometer, with model Nicolet iS10. The experiment was configured with OMNIC software to record measurements in the 4000-400 cm–1 range, with the laboratory atmosphere and the diamond crystal of the ATR accessory as references.”
“UV-Vis spectroscopy was conducted using a Jenway 6850 UVVis spectrophotometer, which provides wavelength resolution within the 300 nm–1100 nm range. The samples were measured in dispersion form, with water used as the reference target, and in thin film form, where a clean substrate served as the reference.”
“For the electrical characterization, current-voltage (I-V) measurements were conducted under dark conditions at room temperature to calculate the DC electrical conductivity (σdc). First, PVA thin films were deposited onto ITO glass substrates. Then, a grid of electrically conductive Ag paint pads was applied and distributed evenly across the sample surface. Each pad was 2 mm in diameter, 1 µm thick, and spaced 5 mm apart. Following this, heat treatment was performed for 30 minutes at 90°C, with a heating ramp of 5°C/min and a cooling ramp of 2°C/min. Current-voltage (I-V) measurements were then carried out using a Keithley 2450 electrometer.”
Comment: In Figure 3b- who depending on who was represented, what A and E represent, should also be referenced, units of measurement
Response: The axis of the plot were modified:
Figure 3. a) UV-Vis absorbance spectra of Cu NPs, b) Tauc variable versus Energy plot for Cu NPs.
Comment: - lines 2017-2019 „connected through the Tauc equation. Thus, the band gap is determined by plotting (ahv)2 against hv (where α is the absorption coefficient, h is Planck's constant, and v is the photon frequency).” - what is the equation or where was it mentioned before in the text of the paper and references
Response: The equations were added in methodology:
“With the optical characterization, the Tauc method was used to estimate the band gap value of the samples. The wavelength and the absorbance were used to calculate the transmittance of the samples with the following equation:
Then, the energy photon (hv) was calculated by:
And also, the absorption coefficient:
Finally, the square of the multiplication of α with hv was calculated. To obtain the band gap, the hv values were plotted in the X-axis, and then the (αhv)2 values were plotted in the Y-axis. The intersection with the X-axis is considered as the band gap value [24]. “
And the corresponding reference was applied:
[24] Makula, P.; Pacia, M.; Macyk, W. How to correctly determine the band gap energy of modifie
semiconductor photocatalysts based on UV-Vis spectra. J. of Phys. Chem. Lett. 2018, 9.
https://doi.org/10.1021/acs.jpclett.8b02892
Comment: line 271, also refer to the number of layers; should be set, rpm or RPM; should it be specified which films were used, films deposited on pure glass or on ITO? Also for Figs. 6, 7
Response: In the following sentence, the information of the layers and substrate was added, and also the term rpm was changed to RPM in this case and at all the paper:
“The UV-Vis absorption spectra of the PVA/Cu films with one, two and three layers deposited at 500, 1000, 1500, and 3000 RPM over PET ITO are presented in Figure 6.”
Response: Also, the substrate was specified in the other figures:
“From the Tauc diagrams (representation shown in the inset of Figure 7), using the UV-visible absorption spectra, band gap values (Eg) of 2.8 eV, 2.6 eV, and 2.8 eV were calculated for films deposited with 1, 2, and 3 layers over PET ITO, respectively, as il-lustrated in Figure 7.”
Comment: line 277, the sentence is repeated „The UV-Vis spectra of the 277 PVA/Cu films deposited at 500, 1000, 1500, and 3000 rpm are shown in Figure 6.”
Response: The sentence ¨ The UV-Vis spectra of the 277 PVA/Cu films deposited at 500, 1000, 1500, and 3000 rpm are shown in Figure 6¨ was eliminated due it was duplicated.
Comment: In 3.3.2. SEM subchapter, „Figures 8-10 show the SEM images of the polymer nanocomposite materials after the incorporation of copper nanoparticles into the polymer matrix.” - It should be clearer on which samples the SEMs were performed; substrate;
Response: The substrate was specified in the sentences of SEM characterization:
“Figures 8-10 show the SEM images of the polymer nanocomposite materials de-posited over PET ITO after the incorporation of copper nanoparticles into the polymer matrix.”
“Micrographs of films deposited over PET ITO in two layers at varying deposition speeds of 500, 1000, and 1500 RPM show the formation of larger aggregates.”
Comment: In Figure 11, A.U. (a. u.); In Figure 12. „3D-AFM Surface topography…” probably 2D to be checked; it seems that the second figure shows a smaller roughness;
Response: In this case we prefer used 3D image, so we will left the report of roughness with the 3D image.
Comment: In Figures 13, 14, another notation of the samples is used „Sample: PVA-5%--Cu-5%_2C”…..also in Table 1, 2, etc. med 1, med 2 without any specification in the text - A consistent notation of samples throughout the text of the paper is recommended
Response: The notation was changed to homogenize all the expressions:
Figure 13. I-V curves for PVA/Cu (5%) nanocomposite films deposited over PET ITO at two layers.
Figure 14. Impedance (Z) evolution for PVA/Cu (5%) nanocomposite films deposited over PET ITO at two layers throughout evaluation frequency.
Figure 15. The series capacitance (Cs) for PVA/Cu (5%) nanocomposite films deposited over PET ITO at two layers.
Figure 16. The calculated dielectric constant (ε) for PVA/Cu (5%) nanocomposite films deposited over PET ITO at two layers.
Comment: line 378, a clearer description, namely: the layers deposited on..... to which electrodes were applied.......were analyzed from the point of view........etc; similar for the characterizations of the Figures 14, 15, 16,
Response: The requested details were concreted in the redaction of the pharagraph.
“As a function of the applied voltage (V), the current (I) flowing through the doped copper PVA thin films was measured. Figure 13 illustrates the I–V characteristics of PVA/Cu (5%) with two layers deposited films for the two deposition speeds (500 and 1000 RPM).”
“Figure 14 presents the impedance (Z) versus frequency (Hz) behavior for a PVA/Cu nanocomposite sample containing two layers of PVA/Cu (5%), synthesized at two dif-ferent deposition speeds (500 RPM and 1000 RPM).”
“Figure 15 illustrates the capacitance (Cs) as a function of frequency (Hz) for PVA/Cu (5%) nanocomposite films with two layers deposited, synthesized at two dif-ferent deposition speeds (500 RPM and 1000 RPM). The curves reveal how capacitance changes across a frequency range from 10 Hz to 1 MHz.”
“Figure 16 shows the dielectric constant (ε) as a function of frequency (Hz) for PVA/Cu (5%) nanocomposite films with two deposited layers, synthesized at two different deposition speeds (500 RPM and 1000 RPM). The dielectric constant (dimensionless) is plotted over a frequency range from 10 Hz to 1 MHz. Again, as in the previous graphs, the behavior of a dielectric material was evident since for all samples the dielectric constant decreases as the frequency increases.”
Comment: In Conclusions, in the first part, a clearer highlighting of the samples obtained, the method of realization and the method used is recommended; „confirming the semiconductive nature of the films” Was the semiconducting nature of the studied layers confirmed only from the UV-VIS study?; „compared to previously reported values” - to be reformulated without reference to the text of the paper;
Response: For the initial suggestion the following pharagraph was complemented:
“This study successfully synthesized and characterized nanostructured films composed of polyvinyl alcohol (PVA) with embedded copper (Cu) nanoparticles. The deposition was homogenous with spin coating method, maintaining these characteristics with 1, 2 and 3 layers; with 500, 1000, 1500 and 3000 RPM of varying deposition.”
For the second suggestion:
“The UV-Vis results demonstrated a decrease in the optical band gap from 4.82 eV for pure PVA to 2.6-2.8 eV for PVA/Cu nanocomposites, values corresponding to a semi-conductive material. The observed reduction in band gap correlates with an increase in copper nanoparticle concentration, suggesting enhanced charge carrier mobility due to improved conductive networks within the matrix.”
“Electrical conductivity measurements indicated significantly enhanced DC con-ductivities, reaching values of 1.20 S/cm for films deposited at 1000 RPM with 5 wt.% Cu. This substantial increase can be attributed to effective nanoparticle dispersion, en-hanced interfacial polarization, and the formation of conductive pathways within the polymer matrix.”
“Dielectric property assessments revealed remarkable performance, particularly at lower frequencies, with dielectric constants reaching values far exceeding those re-ported in existing literature. These electrical properties, in combination with band gaps confirmed the semiconductive nature of the films.”
For the third suggestion, the sentence ¨compared to previously reported values¨ was omitted:
“This substantial increase can be attributed to effective nanoparticle dispersion, enhanced interfacial polarization, and the formation of conductive pathways within the polymer matrix.”

Round 2
Reviewer 1 Report
Comments and Suggestions for Authors
The author should also correct the band gap values of pure PVA in the abstract.
Author Response
Comment: The author should also correct the band gap values of pure PVA in the abstract.
Response: We updated the information of band gap value in the abstract.
“Results indicate a significant reduction in optical band gap, from 4.82 eV for pure PVA to 2.6-2.8 eV in the nanocomposites, alongside enhanced electrical conductivities reaching 1.20 S/cm for films with 5 wt.% Cu.”

Reviewer 2 Report
Comments and Suggestions for Authors
The authors have considerably improwed the work and therefore I recommend its publication.
As a recommendation in Proof., but it is up to the authors to give a reference to equation 4 and use a single abbreviation, a. u. or A. U.
Best regards
Author Response
Comment: As a recommendation in Proof, but it is up to the authors to give a reference to equation 4 and use a single abbreviation, a. u. or A. U.
Response: For the first recommendation, a reference was added in equation 4.
The room temperature conductivity (σRT) was calculated using the following equation [25]:
- Sabanci, S.; Kaya, K.; Goksu, A. Modeling the electrical conductivity value of the model solution. da Acad. Bras. de Cien. 2023, 95. http://dx.doi.org/10.1590/0001-3765202320210062
For the second recommendation, the two plots of DRX using A.U. in the Y-Axis were corrected using a.u., to maintain an uniformity with the UV-Vis plots using a.u. in the Y-Axis.
Response: We accepted a recommendation from the reviewer.
